# Utility of Intermediate Care Units: A Systematic Review Study

**DOI:** 10.3390/healthcare12030296

**Published:** 2024-01-24

**Authors:** Paula López-Jardón, María Cristina Martínez-Fernández, Rubén García-Fernández, Cristian Martín-Vázquez, Rodrigo Verdeal-Dacal

**Affiliations:** 1Asturias Medical Centre, 33193 Asturias, Spain; 2SALBIS Research Group, Faculty of Health Sciences, Department of Nursing and Physiotherapy, Campus de Ponferrada, Universidad de León, 24401 León, Spain; rgarcf@unileon.es; 3Nursing Research, Innovation and Development Centre of Lisbon (CIDNUR), Nursing School of Lisbon, 1600-190 Lisbon, Portugal; 4Department of Nursing and Physiotherapy, Campus de Ponferrada, Universidad de León, 24401 León, Spain; cmartv@unileon.es; 5Servicio de Medicina Interna, Hospital Público de Valdeorras, 32300 Ourense, Spain

**Keywords:** intermediate care units, benefits, intermediate care facilities, effectiveness, intensive medicine, patient comfort

## Abstract

Intermediate care units (IMCUs) have become increasingly important in the care of critical and semi-critical patients, particularly during the COVID-19 pandemic. However, there is still no clear definition of their structural characteristics, specialties, types of patients, and the benefits they provide. The aim of this work is to describe the current state of implementation and operation of IMCUs in hospitals and patient care. To achieve this goal, a systematic review was conducted in the Web of Science, Scopus and CINAHL databases, along with a hand search. The research yielded 419 documents, of which 26 were included in this review after applying inclusion and exclusion criteria. The results were highly diverse and were categorized based on the following topics: material resources, human resources, continuity of care, and patient benefits. Despite the different objectives outlined in the studies, all of them demonstrate the numerous benefits provided by an IMCU, along with the increased relevance of this type of unit in recent years. Therefore, this systematic review highlights the benefits of IMCUs in the care of critical patients, as well as the role of health workers in these units.

## 1. Introduction

The COVID-19 pandemic has adversely affected the functioning of the health system [1]. Intermediate care units (IMCUs) have emerged as a crucial resource in the wake of the COVID-19 pandemic and are a fundamental resource [2]. They provide continuous care to critically ill patients who require constant monitoring of their vital signs and/or frequent nursing interventions [3], ensuring high-quality treatment. An additional benefit of its implementation was the limited availability of intensive care unit (ICU) beds, leading to premature discharge and consequent negative effects on the patient, such as increased mortality rates, re-admissions, and prolonged hospitalization [4]. 

Generally, hospitals offer two types of care: inpatient units and ICU [3]. However, certain patients have clinical characteristics that require a higher level of care than standard inpatient units, but do not necessitate ICU care [5]. These patients are considered semi-critical and are potential candidates for IMCU beds. As a result, the American College of Critical Care Medicine (ACCCM) has established clinical guidelines and admission criteria that are widely accepted in such units. However, these recommendations must be adapted to the specific needs and environment of each hospital, allowing for some flexibility [3,6]. In accordance with these guidelines, IMCUs should be overseen by a physician with experience in intensive care medicine [6]. According to ACCCM admission criteria for an IMCU, patients must have stable cardiovascular conditions (such as uncomplicated AMI, ACS, arrhythmias, and pacemaker implantation) and respiratory conditions (including failure and ventilated patients) to ensure comprehensive care for a wide range of cases covering cardiac, respiratory, and neurological areas [7].

Additionally, specific criteria exist for determining which patients are transferred to an IMCU [3,6], including those recovering from ICU care, patients from the ward or emergency room experiencing poor outcomes requiring a high level of nursing care, or seriously ill patients. Difficult cases that require non-invasive techniques will receive attention, and patients requiring adjustments in fluid therapy will be seen as well. 

However, while considering an official perspective, there is a still a need for clear definition regarding the structural attributes, specialties, and patient categories of IMCUs. Nevertheless, different scientific societies have proposed various models for IMCUs, which vary based on the proposer: parallel model and stand-alone model. Another model that combine both units is the integration model. For example, the German Interdisciplinary Association for Intensive Care and Emergency Medicine (DIVI) recommends including an IMCU either within an ICU or as an independent IMCU [7]. Intensive care and intermediate care patients are treated together to increase workforce flexibility in terms of nurse-to-patient ratio and to allow for adaptable treatment adjustments to meet patient requirements. Transfers between units are possible. Furthermore, duplicating monitoring equipment in both the ICU and IMCU may be unnecessary, resulting in significant cost reductions [8,9]. In a parallel model, the ICU and IMCU are situated in distinct regions while remaining adjacent to each other and having equal access to resources. Units following this model will enjoy greater flexibility with nursing staff exchanges and a continuous presence of an intensivist physician. In case of patient transfer between units, information loss will be minimal, guaranteeing outstanding treatment and care continuity. Additionally, the ICU personnel will receive immediate assistance from other ICU staff in the event of medical emergencies [8,9]. Last, a stand-alone IMCU is an autonomous unit in terms of space, organization, and personnel. This model is a suitable solution when there are structural constraints in the ICU. Moreover, it can serve as a treatment unit for hospitals without an ICU if it does not replace the existing ICU. However, this model may impact the continuity of care (transfer of patients), necessitate a complete set of its own resources and staff, and significantly reduce staff flexibility [8,9].

It can be stated that IMCUs are hospital areas provided with sufficient human and material resources to ensure monitoring and care at a level lower than that of ICUs, but much higher than that of conventional hospitalization areas [10]. Restructuring of critical and semi-critical care units can be efficient if it minimizes costs, reduces the consumption of time and resources, without compromising the quality of care for critical and semi-critical patients [11]. Interest in IMCUs has increased in both Europe and the United States. A systematic review has highlighted the positive effects of IMCUs on health systems, including reduced waiting times, improved synchronization of different medical resources, and flexible patient flow [12]. However, it also highlights the challenge of identifying this area of expertise as there is no shared understanding of this type of unit [12].

Given the limited understanding of the global perspective of IMCUs, this work aims to primarily describe the current state of implementation and operation of IMCU both to a hospital and to patient care. Additionally, this study seeks to identify the material and human resources available in an ICU, define the patient profile that can benefit from this intermediate care unit, and analyze the role played by ICUs during the COVID-19 pandemic.

## 2. Materials and Methods

A systematic review of the literature was conducted. This review was based on the Preferred Reporting Items for Systematic Reviews and Meta-Analyses (PRISMA) methodology [13]. A protocol was registered with the Open Software Foundation (OSF) (https://osf.io/y2veq/, accessed on 27 December 2023, registration id: osf.io/y2veq/).

### 2.1. Search Strategy

The SPIDER tool (Sample, Phenomenon of Interest, Design, Evaluation, and Research strategy) [14] was used to guide the information search process in formulating the following research question. What is the current situation of implementation and operation of Intermediate Care Units (IMCUs) in the global healthcare context, and how does this affect the availability of material and human resources? (Table 1).

A search of the Web of Science, Scopus, and CINAHL databases was carried out during October–November 2023. Additionally, relevant articles in Intensive and Critical Care Medicine and Nursing were selected with a hand search of major journals, and gray literature was also searched. 

### 2.2. Search 

To focus the search for information, the following Health Sciences (DeCS) and Medical Subject Headings (MeSH) terms were used: “intermediate care units”, “benefits”, “intermediate care facilities”, “effectiveness”, “intensive medicine”, and “patient comfort”. At the same time, to narrow down the search further, Boolean AND was used.

In each of the databases, several combinations of the above descriptors were made using the Boolean operator AND until the most appropriate combination for the objectives of this study was obtained. The following search equations were used to retrieve the studies: “Intermediate care units” AND benefits, “intermediate care units” AND “intermediate care facilities”, “Intermediate care units” AND “intensive medicine”, “intermediate care units” AND effectiveness, and “intermediate care units” AND “patient comfort”.

### 2.3. Study Selection

All retrieved studies were imported into the Mendeley Desktop bibliographic manager, with duplicate studies being removed. The article selection process was conducted using the following inclusion criteria: articles in English or Spanish and articles with the keyword “intermediate care units” in the title.

Additionally, we excluded articles that were systematic reviews, studies with a pathology-specific focus, and those that referred to the intermediate care unit as an extension unit for chronic patients or home care, rather than as a center for semi-critical patients. No studies were excluded based on their date of publication as this is a topic that has been underexplored, which offers a broader perspective on the relevant subject matter. The studies underwent independent screening by two reviewers, PLJ and MCMF. Any conflicts were resolved through discussion.

## 3. Results

A total of 419 articles were retrieved through the search. The process for selecting articles is outlined in Figure 1, showing the PRISMA diagram. Finally, the systematic review included a total of 19 studies, of which 4 were identified through handsearching.

When considering the country of origin of the included studies, it is noteworthy that the majority, 36.4%, were published in Spain. Following Spain, the distribution was as follows: U.S.A. (15.4%), Belgium, Italy, Netherlands, and Germany (7.7%, respectively), and Canada, China, Colombia, Brazil, and Turkey (with 3.8%, respectively). This could be attributed to the characteristics of the Spanish healthcare system and the specific interest in investigating the potential of an IMCU. After conducting a comprehensive analysis of the articles, we employed a thematic approach to evaluate the data in reference to the aims and outcomes outlined in Table 2. Our analysis yielded significant insights related to four primary elements: material resources, human resources, continuity of care, and patient benefits.

Material resources

IMCUs are viewed as a substitute to traditional hospitalization, allowing for an expansion in the number of beds allocated for critically ill patients [3,29]. It is generally suggested that the number of beds in a unit should not surpass 10–12, as this higher total could prove problematic to manage [7]. Moreover, IMCUs require both invasive and non-invasive monitoring, which can result in additional costs and is often not cost-effective [15]. The use of IMCUs can reduce the burden on ICUs by admitting patients with lower acuity, but it has been found that ICUs have more complex pathologies and higher workloads [25].

Human resources.

Providing a high-quality clinical environment necessitates a multidisciplinary healthcare team [18]. Although IMCUs may be staffed with fewer nurses, research shows that they require the same level of training as ICU nurses [15]. The German Interdisciplinary Association of Intensive Care and Emergency Medicine (DIVI) recommends that at least 20% of IMCU staff possess intensive care knowledge [7], as it will enhance confidence and training, thereby improving patient care quality [20]. According to various critical care associations, maintaining a nurse-to-patient ratio of 1:2 or 1:3 is essential to guarantee quality care [7,17,27]. Respiratory intermediate care units result in a decreased number of hospitalization days compared to patients in internal medicine units, thereby directly affecting the resources and organization of the hospital [22].

Continuity of care.

Reducing the “care gap” between an ICU and an inpatient unit is the primary goal of implementing IMCUs [10]. Additionally, establishing these units decreases the occurrence of premature discharges [3], eliminates the need for transfers to regional hospitals lacking ICUs [18], and minimizes the length of stay in the ICU while still maintaining proper levels of surveillance and monitoring [23] without adversely affecting patient outcomes [17]. Intermediate care units have proven to be a critical asset in managing extremely complicated patients during the COVID-19 outbreak [32,36]. 

Patient benefits.

The environmental conditions within an IMCU play a crucial role in a patient’s clinical progress. Managing alarms and promoting a culture of silence can reduce noise levels and thus improve patient outcomes [24]. Furthermore, the presence of family members throughout the patient’s stay within the unit provides continuous support and offers psychological benefits to the patient [3]. During the COVID-19 pandemic, respiratory IMCUs prevented almost 50% of critically ill patients [37] from being admitted to ICUs. They also served as a multidisciplinary support for managing COVID-19 patients in critical conditions who required respiratory support and non-invasive monitoring [31].

## 4. Discussion

This systematic review aimed to describe the current state of implementation and operation of IMCUs. It was found that they provide improvements in effectiveness and cost, resource use, and continuity of care for patients who appear to have little chance of complications in the evolution of their disease. The key challenge is how to efficiently allocate resources to meet the needs of non-critical patients who still require potential access to critical care in the future.

Material resources

IMCUs offer a primary benefit as an alternative to ICUs, which increased bed availability for patients with higher levels of severity [3,25], and they also enable ICUs to accommodate more complex patients [25]. Patient mortality risk could be a useful metric for rationalizing and optimizing patient admissions, particularly given the limited capacity of higher levels of care in hospitals [29]. A multicenter study conducted in the US by JL. Vincent and GD. Rubenfeld found that 20% of UCI revenue was being used unnecessary for low-severity critical care, which is clinically inappropriate [23]. Vincent and Burchardi suggest that a minority of severely ill patients utilize a disproportionate amount of ICU resources. Therefore, reducing the number of these patients may have a relatively insignificant impact on the overall expenses of an ICU [15]. In this context, a study suggests an advantage of an IMCU that is physically and administratively independent of the ICU. Hospitals with an IMCU have lower mortality rates than those without one [21]. Another study highlights that a closed IMCU model is also considered desirable [35]. The implementation of an IMCU in a hospital has been observed to lead to increased costs. However, this increase is primarily due to surgical admissions and extended stays of patients within the ICU service. Underutilization of the IMCU may contribute to increased costs as nursing staff are paid without patients to attend to [19]. During the COVID-19 pandemic, ICUMs were a fundamental resource in Spain due to their ability to provide a higher nurse-to-patient and doctor-to-patient ratio compared to traditional critical care units. This was particularly important given the increased demand for critical care. Additionally, they were used to transfer ICU patients who required high-flow nasal cannulae [33].

There is no unanimous agreement on the appropriate IMCU bed capacity, according to scientific evidence. Several hospitals determine their needs using ICU studies (Bridgman formula), different specialization percentages, emergency department pressure, surgical waiting lists, possibility of ICU drainage, as well as the number of refused transfers from other centers [17]. However, in Germany and similar countries, DIVI (the German Interdisciplinary Association for Intensive Care and Emergency Medicine) suggests units of 10–12 beds due to the difficulty of managing very large units of 22–28 beds [7]. Furthermore, while IMCUs have less invasive monitoring equipment, they also require invasive monitoring instruments for potential emergencies, resulting in additional expenses [15].

Human resources

One of the primary economic benefits of IMCUs is the reduction in staff numbers. Nonetheless, recent research indicates that whilst the number of nurses may be smaller, they must receive the same level of training as their ICU counterparts [15]. Additionally, as a multidisciplinary team, there should be space for the involvement of other specialists, including intensivists, anesthesiologists, or experts in the critical management of patients [18].

Ensuring a high-quality clinical environment necessitates critical thinking on the part of nursing professionals. A study conducted by Andrew D. Harding et al. at Morton Hospital in Massachusetts demonstrated that enhancing the education and confidence of nursing staff resulted in improvements to patient care quality [20]. This view is corroborated by the German Interdisciplinary Association for Intensive Care and Emergency Medicine (DIVI), which suggests that a minimum of 20% of nurses on the ICU team must receive intensive care training [7]. However, Vincent and Burchardi argue for the importance of not segregating patients based on their severity, which enables staff to maintain their level of attention and interest, and prevents patients from feeling inferior to their colleagues in the ICU [15]. Castillo and colleagues propose the potential for healthcare personnel to adjust the tempo of care delivery by alternating between critically ill patient care and intermediate level care [3].

It is evident that staffing in the IMCU is contingent on patient type, required therapeutic effort, and care team composition. The AACN recommends a nurse-patient ratio of 1:3 or 1:4 [17]. Some countries, like Great Britain, have implemented the 1:3 ratio model, while others, such as Switzerland, advocate for a flexible ratio based on patient severity [7]. These high ratios of nurses to patients permit the admission of patients with more complex requirements, resulting in a greater workload for the nursing staff [27].

Continuity of care

The purpose of IMCUs is to decrease the gap in care between the ICU and hospital wards and ensure uninterrupted care [10]. According to various authors, such as F. Castillo et al. [3], these units reduce the number of premature discharges by allowing patients with decreasing care needs to be transferred to the IMCU for gradual care, thereby avoiding the risk of involuntary and/or inappropriate discharges. Moreover, in regional hospitals lacking an ICU, such as the Hospital Valle del Nalón in Asturias (Spain), these units prevent unnecessary transfers and optimize those that are necessary in the end [18].

Other recent studies, such as the one conducted by JL. Vincent and GD. Rubenfeld, have demonstrated that 20–30% of patients admitted to an ICU are there for less than 24 h for routine surveillance and monitoring [23]. Consequently, the creation of IMCUs would decrease the average ICU stay for patients without negatively affecting their clinical outcomes [17]. If these units were not available, patients of this type would receive treatment directly on an inpatient ward, with care levels significantly lower than what is required, as noted by S. Heili-Frades et al. [28]. However, authors such as J.L. Vincent and H. Burchardi argue that segregating patients into “intensive” and “intermediate” categories could be interpreted as diminishing the importance of the patient as an individual person [15]. During the COVID-19 pandemic, the demand for ICU beds exceeded the supply, and this data are particularly relevant, given the importance of admission criteria. Prior to admission ICU, it is essential to manage critically ill patients for the application of specific drugs or procedures that may impact their outcome during the pandemic period [38]. According to Galdeano et al.’s study, 70% of COVID-19 patients were treated in respiratory IMCUs, highlighting their efficacy as support units [32]. 

Patient benefits

The main objective of an IMCU is to improve the patient’s environment to allow a better evolution of their pathology. To this end, the environmental dimension in which they find themselves is essential since satisfaction and comfort have a significant impact on the prognosis of their disease. For example, the management of alarms and the promotion of a culture of silence can significantly reduce patient discomfort by reducing the noise in the environment [24].

Objective criteria need to be established for the admission of patients to ICUs and IMCUs. Failure to do so can have negative consequences. For example, admitting a critically ill patient to an IMCU first and then transferring them to the ICU after their condition worsens has been shown to increase mortality [30]. Allocating elderly general surgery patients who do not require organ support therapy in an ICU to an IMCU instead of standard wards can significantly reduce 8.7% of predefined life-threatening postoperative complications [34], along with decreasing patient treatment costs, in this study [16,34]. In addition, F. Castillo et al. show that the stay in an IMCU can be considered as a psychological advantage for the patient. It allows a gradual change from a level of maximum assistance to a lower level, with the possibility of being accompanied by the family at all times, thus increasing their level of comfort [3]. However, it should be noted that IMCUs often treat patients with multiple comorbidities and occasionally with indications for DNR [33].

The crucial role of respiratory IMCUs during the COVID-19 pandemic, which revealed the shortage of ICU beds in the Spanish healthcare system, should be emphasized [32]. As demonstrated by G. Suarez-Cuartin et al., such units enabled the treatment of patients who needed constant monitoring but did not require invasive mechanical ventilation [31], thus preventing almost half of critically ill patients from being admitted to an ICU [37]. Therefore, IMCUs provide safe environments in which more complex care can be administered and require constant evaluation by professionals, and where low mortality rates have been observed compared to critical care [31]. Furthermore, the multidisciplinary teams in the IMCUs have successfully reduced the burden on ICUs when treating patients with COVID-19 [36]. This is aligned with a prior study that highlights the benefits of IMCUs in diminishing mortality, the necessity for ICU admission, and hospital stay duration for patients with intricate respiratory ailments [22]. Furthermore, an IMCU is advantageous for enhancing the management of patients who require complex care but do not need ICU admission.

Several limitations were found in this systematic review. The foremost among them is the challenge of comparing studies conducted in various existing IMCUs due to differences in objectives. In addition, the limited amount of available literature has necessitated the inclusion of articles from a wider timeframe. The body of knowledge on intensive care medical units is complex and varies from country to country. Published studies are often specific to certain interventions, making it difficult to fully understand the potential of IMCUs.

## 5. Conclusions

The effectiveness of using IMCUs for the treatment of critically ill patients who, due to their pathologies, are not suitable for or cannot be admitted to a traditional ICU has been established. The characteristics of IMCUs enable high-quality care provision, with emphasis on the patient’s wellbeing and a decrease in the occupancy rate of regular ICU beds. Furthermore, the COVID-19 pandemic highlighted the importance of this unit as it provided necessary care to patients who may not have received the appropriate attention otherwise. In response to the need for beds for more critical patients, however, IMCUs have become increasingly involved in the management of semi-critical patients. Research has demonstrated that while IMCUs may differ in certain aspects, they share similarities in terms of patient profiles and a higher patient-to-staff ratio than traditional ICUs. Additionally, IMCUs are staffed with professionals who are equipped to manage complex patients with diverse pathologies, providing an advantage over standard hospital wards. However, further studies are required to determine the fundamental characteristics of this type of unit, as the current evidence remains scarce.

As a future line of research, this study reinforces the significance of IMCUs in the care of critically ill patients in hospitals that already have an ICU. For hospitals that have not yet implemented IMCUs, this study provides compelling reasons to consider changes in resource allocation or nursing staff training.

## Figures and Tables

**Figure 1 healthcare-12-00296-f001:**
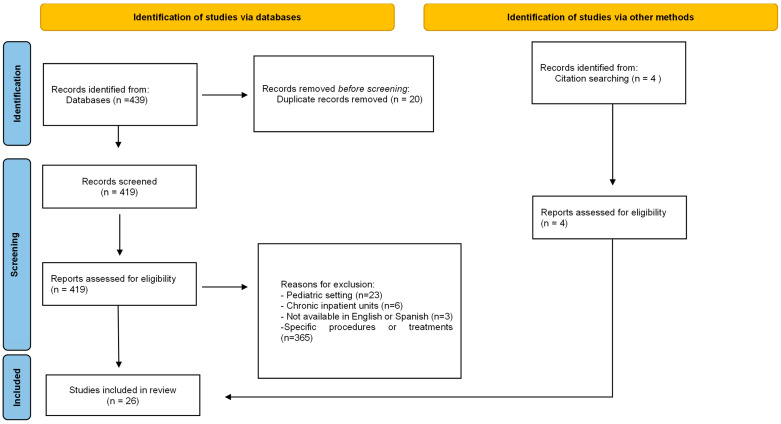
Prisma flowchart.

**Table 1 healthcare-12-00296-t001:** SPIDER strategy.

Sample	Intermediate Care Units
Phenomenon of Interest	Implementation and current situation of IMCU
Design	Systematic review
Evaluation	Description of the current context
Research type	Qualitative

**Table 2 healthcare-12-00296-t002:** Results of the systematic review.

Author, Year and Country	Aim of the Study	Relevant Results
J.L. Vincent and H. Burchardi (1999, Belgium) [15]	To explore the advantages and disadvantages of IMCUs and present suggestions regarding their integration into hospital settings.	- Arguments against establishing an independent IMCU include a lack of significant cost reduction, the need for highly qualified nurses, inability to avoid ICU overload, and lack of continuity of care. - The benefits of integrating an IMCU in an ICU are significant since it optimizes resource organization and use.
P. Senaratne et al. (1999, Canada) [16]	To determine whether it is feasible to eliminate the period of hospitalization after experiencing acute myocardial infarction and release patients directly from a coronary intermediate care unit.	- Discharge directly from a Coronary IMCU is feasible in most patients admitted with AMI. Discharge does not increase morbidity and mortality six weeks after discharge and appears to reduce costs.
G. Martínez Estalella (2002, Spain) [17]	Determine the necessary characteristics before implementing an IMCU in a hospital.	- Cost of an ICU: the care resources needed to meet the criteria of effectiveness, efficiency, and equity. - Formula for determining the number of beds needed. - Main benefits of implementing an ICU.
F. Castillo et al. (2007, Spain) [3]	Collection of current knowledge about IMCUs for their subsequent implementation and improvement of future units in a hospital environment.	- Cost containment strategies. - Benefits of the IMCU to the level of care and effectiveness of the unit and other hospital services. - Flow of inpatients and discharged patients between an IMCU and other units.
A. Heras et al. (2007, Spain) [10]	To analyze the impact of opening an IMCU at a referral center.	- Origin of patients admitted to an IMCU. - Care benefits provided by the unit. - Increase hospital care capacity without affecting overall mortality rates.
J. Alfonso-Megido et al. (2007, Spain) [18]	To present an open IMCU model implemented at the Hospital of Asturias without an ICU, avoiding unnecessary transfers or optimizing those that are necessary.	- Optimal Role of IMCUs in regional hospitals without ICUs: evaluating care, reducing costs, enhancing continuity of care, and improving patient and family satisfaction. - Decrease in the number of transfers was observed. - Personal resources used in this unit.
B. Solberg et al. (2008, The Netherlands) [19]	Determine whether the introduction of an IMCU reduces total hospital and special care costs.	- The total hospital costs for each patient increased after the introduction of an IMCU. This increase is not due to the unit itself but to the specific characteristics of these patients, with a more surgical profile and high therapeutic requirements.
Andrew D. Harding et al. (2009, USA) [20]	To explain the utilization of an IMCU as a tactic to replace skilled critical care nurses departing from the profession.	- Reduce the daily cost of patient care. - Improved continuity of care. - Benefits to critical care nurse education.
M. Capuzzo et al. (2014, Italy) [21]	European study to determine whether adults managed in hospitals with an ICU and IMCU have lower mortality than those managed in hospitals without an IMCU.	- Most ICUs were located in a hospital equipped with an independent IMCU. - There is a link between the presence of an IMCU and a reduction in mortality rates among patients requiring intensive care in the ICU.
M. Confalonieri et al. (2015, Italy) [22]	Effects of a respiratory IMCU on the improvement of patients with acute respiratory failure or exacerbations of chronic obstructive pulmonary disease or community-acquired pneumonia.	- Reduced in-hospital mortality and need for ICU admission in patients with respiratory pathology. - In comparison to both emergency units and internal medicine, mortality rates have decreased. - Decrease in the time needed to apply non-invasive mechanical ventilation and specific medications.
JL. Vincent and GD. Rubenfeld (2015, Belgium) [23]	To examine the available data on the flexibility and efficiency of an IMCU model within an ICU.	- Arguments for and against IMCUs. - Effects on costs and results. - Efficiency of this type of units.
A. González Gómez et al. (2017, Colombia) [24]	Associate sociodemographic factors with the dimensions of comfort (physical, social, psychospiritual, and environmental) that affect patients hospitalized in IMCUs.	The importance attributed by patients to various factors is contingent upon their physical condition, socioeconomic and educational status, as well as social and environmental circumstances.
C. Waydhas et al. (2018, Germany) [7]	Describe the recommendations of the German Interdisciplinary Association for Intensive Care and Emergency Medicine regarding staffing, capacity, equipment, and structure of IMCUs.	- Organization models for ICUs. - Scientific evidence on the number of beds in relation to the structural model of the unit. - Available material and personnel resources.
U. Hamsen (2018, Germany) [25]	Determine if the provision of an IMCU for the general ICU population leads to a more suitable distribution of patients.	- ICUs with high IMCU usage tend to have younger patients with more severe illnesses and a higher workload. - ICUs with low IMCU usage discharged a lower percentage of patients compared to those with high IMCU usage.
B. Wendlant et al. (2018, USA) [26]	To identify patterns of IMCU utilization in US hospitals.	- ICUs are mixed units that treat medical, cardiac, and surgical patients. - In 21% of the IMCU, intensivists managed the care, while in 36% of the units, the care was not managed by intensivists.
J. DJ Plate et al. (2019, The Netherlands) [27]	Determine whether a hospital’s IMCU decreases healthcare expenses.	- A comparison of costs per patient between the IMCU and ICU demonstrates economic benefits of the IMCU. - Adequate and consistent triage is crucial to optimizing the IMCU’s potential. - The IMCU yields cost savings of EUR 1,558,965 annually.
S. Heili-Frades et al. (2019, Spain) [28]	To perform a cost analysis of a respiratory IMCU in a general hospital to determine the annual expenses linked to its complexity and its potential effectiveness in terms of avoided costs.	- Considerable economic savings can be achieved through avoidance of lengthy hospital stays and improved allocation of resources. - Despite the complexity of patients, low mortality rates were observed. - A respiratory IMCU reduces costs and maintains a low mortality rate. - Provide intricate care to patients with complex respiratory conditions, thereby avoiding unnecessary or prolonged stays in the ICU.
A. Meisami et al. (2019, USA) [29]	To develop a methodology for selectively admitting patients to an ICU and IMCU, which incorporates patient health risk metrics and considers the potential for congestion.	- The optimized methodology led to a 37% increase in average weekly admissions to ICUs and a 12% increase in average weekly admissions to IMCUs, with minimal blockage. - The optimized model indicated a reduction in the risk levels required for admission.
JGR Ramos et al. (2021, Brazil) [30]	To examine the frequency of escalated transfers from an IMCU to an ICU among patients admitted on an emergency basis and to evaluate this effect.	- The number of patients transferred from an IMCU to an ICU was higher than the number admitted to an ICU alone. - Transfers were associated with increased mortality rates. - Transfers were primarily due to progressive deterioration of patient’s original condition.
G. Suarez-Cuartin et al. (2021, Spain) [31]	To evaluate the outcomes of COVID-19 patients requiring non-invasive monitoring and respiratory assistance admitted into an IMCU and to determine the clinical factors that may contribute to these outcomes.	- IMCUs alleviate the strain on ICU resources for COVID-19-positive patients who need respiratory support and non-invasive monitoring. - Protective and risk factors of COVID-19 positive patients in the IMCU setting.
M. Galdeano Lozano et al. (2021, Spain) [32]	Assess the effectiveness of a respiratory IMCU located in a tertiary hospital, including the epidemiological and clinical features, as well as the mortality rate of COVID-19 patients.	- Importance of respiratory IMCUs during the COVID-19 pandemic to facilitate the management of high patient volume. - The management of high patient volume and reduction of ICU stays and income loss.
M. Matute-Villacís et al. (2018, Spain) [33]	Describe the establishment of two respiratory IMCUs to care for patients during the COVID-19 pandemic, their characteristics, and patient outcomes.	- The functional organization of these units reduced the risk of virus transmission among professionals. - Patients transferred from non-ICU services were older, had more comorbidities, and lower BMIs, resulting in higher mortality rates. Many of these patients had a DNR order.
L. Wang et al. (2021, China) [34]	To investigate retrospectively the impact of an IMCU on the costs and outcomes of general surgery patients in the elderly population.	- The incidence of life-threatening postoperative complications reduces by 8.7%. - There was a reduction in the cost of patient treatment.
D. Hager et al. (2022, USA) [35]	Identifying the optimal organizational model for managing staff in an IMCU.	- The closed ICU staffing model was deemed the most effective. - Its advantages include increased nursing satisfaction, smoother patient transfers between different levels of care, and reduced costs.
H. Bülbül et al. (2023, Turkey) [36]	To determine survival and parameters predictive of mortality in patients admitted to an IMCU.	- Most patients had a pulmonary pathology when admitted. - The implementation of multidisciplinary patient management teams for respiratory pathology and monitoring in COVID-19 patients reduced ICU occupancy. - Early transfer of patients to the ICU was found to improve survival rates.
C. Caballero-Erasoa et al. (2022, Spain) [37]	Conduct a prospective study of all respiratory IMCUs in Spain to investigate their role in pulmonology throughout the COVID-19 pandemic. Explore the expansion of respiratory IMCUs, the challenges they have encountered in care provision, and the clinical outcomes they have attained.	- Increased respiratory care for critically ill patients with COVID-19 and its significance in treatment. - The ability to adapt and respond to Pneumology services during the pandemic is crucial. - It is important to establish a respiratory network to provide unified critical care in all regions.

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
