# Peer review of "Utility of Intermediate Care Units: A Systematic Review Study"

_healthcare, 2024, doi:10.3390/healthcare12030296_

Round 1

Reviewer 1 Report

Comments and Suggestions for Authors

The authors do a systematic review on the implementation of the imcu. I think the paper is well-written and the topic is relevant. However, I think that the authors should perhaps do more effort in the search of documents.

It seems that the question performed, “Does the implementation of an Intermediate Care Unit bring benefits to hospitals and patients?”  does not fully match with the target reported “describe the current state of implementation and operation of IMCU both to 86 a hospital and to patient care”. To describe the current state is not to check the benefits.

Perhaps the search should include more dataset, for instance, google.scholar, med.archives, other non-academic sources.

Again, why the words ‘benefits’ and ‘patient comfort’ were included? Those are not in the target and could bias the search. For instance, you are not including terms such as ‘cost’ or ‘complexity’.

What was the reason of ‘were unavailable for full-text access’. I think, in general, this should not be a good reason. If I do not have access to the papers I cannot do a review. Other thing is that the document is not available for no one. If your institution is not supporting the journal is not a reason. Why you exclude previous systematic reviews? I understand that if there are previous systematic reviews these should be commented in the introduction.

It is surprising that 42% of the documents are from Spain (country of the authors) and just one paper from USA. The authors should explain this point.

Since not objective data are reported, I think the conclusion can be strongly for the role of the authors of the papers. Could you report it?

Line 65. possible.Furthermore

Author Response

Thank you for your feedback. Please see the attatchment .

Reviewer 2 Report

Comments and Suggestions for Authors

Thank you for the opportunity to read this interesting work. I am asking the authors to implement the following corrections:

1) Please complete the methodology in the abstract (e.g. how many articles were originally found and what were the inclusion criteria)

2) The literature contains only 8 works that can be considered current (not older than 5 years). Please complete the bibliography with the latest articles relating to the main topic:

a) Leszczyński PK, Sobolewska P, Muraczyńska B, Gryz P, Kwapisz A. Impact of COVID-19 Pandemic on Quality of Health Services Provided by Emergency Medical Services and Emergency Departments in the Opinion of Patients: Pilot Study. Int J Environ Res Public Health. 2022; 19: 1232. DOI: 10.3390/ijerph19031232

b) Ceylan I, Karakoç E, Güler G, GöktaÅŸ SY, Ökmen K, Sayan HE. The impact of early antibiotic initiation on ICU mortality in severe COVID-19 patients. Crit. Care Innov. 2023; 6(2): 1-9. DOI: 10.32114/CCI.2023.6.2.1.9

3) Why did the authors add a self-search in the journals "Intensive Care" and "Critical Care Medicine and Nursing"? Adding independent searches outside databases (WoS, SCOPUS, Index Copernicus, PubMed, etc.) raises methodological doubts.

4) What was the problem with access to the full text of rejected articles? If it was a matter of payment for access, why didn't the authors use subscriptions for scientific units?

5) I believe that the publication date should be verified during the PRISMA review, as outdated information from many years ago (e.g. from 1999) may disturb the contemporary concept of ICU work.

6) In Figure 1, complete the number N in the "Reasons for exclusion" box

Author Response

Thank you for your feedback. Please see the attachment

Round 2

Reviewer 1 Report

Comments and Suggestions for Authors

I think the authors have done a very good job responding to my previous questions and addressing the reviewers suggestions.

I agree with the comment 'we do not fully understand the USA handling'... I live in the USA and work on health environments and do not understand the system.

As a note, as a person who grew up 50 meters from the 'Hospital Valle del Nalon' (aka, hospital de villa), I am very glad to read about this institution in an Academic paper.